

# Species diversity and drivers of arbuscular mycorrhizal fungal communities in a semi-arid mountain in China

He Zhao[*], Xuanzhen Li[*], Zhiming Zhang, Yong Zhao, Jiantao Yang and Yiwei Zhu

College of Forestry, Henan Agricultural University, Zhengzhou, China
[*] These authors contributed equally to this work.

## ABSTRACT

Arbuscular mycorrhizal fungi (AMF) play an essential role in complex ecosystems. However, the species diversity and composition of AMF communities remain unclear in semi-arid mountains. Further, it is not well understood if the characteristics of AMF community assemblies differ for different habitat types, e.g., agricultural arable land, artificial forest land, natural grassland, and bush/wood land. Here, using the high-throughput technology by Illumina sequencing on the MiSeq platform, we explored the species diversity and composition of soil AMF communities among different habitat types in a semi-arid mountain (Taihang Mountain, Mid-western region of China). Then, we analyzed the effect of nutrient composition and soil texture on AMF community assembly. Our results showed that members of the *Glomus* genera were predominated in all soil types. The distance-based redundancy analysis indicated that the content of water, available phosphorus, and available potassium were the most crucial geochemical factors that significantly affected AMF communities ($p < 0.05$). The analysis of the soil texture confirmed that AMF diversity was negatively correlated with soil clay content. The comparison of AMF diversity among the various habitat types revealed that the artificial forest land had the lowest AMF diversity in comparison with other land types. Our findings suggest that there were differences in species diversity and composition of soil AMF communities among different habitat types. These findings shed new light on the characteristics of community structure and drivers of community assembly in AMF in semi-arid mountains, and point to the potential importance of different habitat types on AMF communities.

Corresponding author
Yong Zhao, zhaoyonghnnd@163.com

## INTRODUCTION

Arbuscular mycorrhizal fungi (AMF) play a high-value role for ecosystem restoration and sustainability (*Herder et al., 2010*; *Sanders, 2010*; *Verbruggen et al., 2012*). The majority of land plant species have the potential ability to form symbiotic relationships with AMF, which can significantly enhance plant growth (*Lekberg & Koide, 2005*), improve soil structure (*Piotrowski et al., 2004*; *Caravaca et al., 2006*; *Wilson et al., 2009*), and contribute to plant resistance to environmental stress (*Benjamina, Karl & Johnn, 2009*; *Balliu, Sallaku & Rewald, 2015*). AMF also can maintain ecosystems stability and promote ecosystem

development (*Larsen, Williams & Kremen, 2005*; *Fuhrman, 2009*; *Rosindell, Hubbell & Etienne, 2011*). Therefore, to explore the ecological environment in diverse regions, understanding AMF diversity and biogeography will be of primary importance (*Fitter, 2005*; *Chaudhry et al., 2012*).

In recent years, many studies have reported the AMF community composition in different environmental conditions (*Öpik et al., 2006*; *Wubet et al., 2006*; *Heijden & Scheublin, 2007*; *Lee, Lee & Young, 2008*; *Krüger et al., 2009*). Scholars have argued that the composition of AMF communities will vary along the gradients of land-use intensity under the same climatic conditions and region of agricultural ecosystems (*Dumbrell et al., 2010*; *Fritz et al., 2010*; *Lekberg et al., 2011*; *Mirás-Avalos et al., 2011*; *Meadow & Zabinski, 2012*). Also, several papers have confirmed that the AMF distributions are caused by their ability to tolerate high nutrient concentrations in different habitat types (*Porras-Alfaro et al., 2007*; *Egertonwarburton, Johnson & Allen, 2008*; *Thomson, Robson & Abbott, 2010*). Meanwhile, through the investigation of natural or agricultural habitats, scholars have shown that a high diversity of rhizosphere AMF was found in natural habitat (*Öpik et al., 2008*; *Bonfim et al., 2016*), and the AMF communities inhabiting plant roots tended to have a lower diversity in agricultural ecosystems (*Daniell et al., 2001*; *Alguacil et al., 2011*; *Schnoor et al., 2011*; *Bainard et al., 2015*). However, most of the previous research works focused on single ecosystems (*Helgason et al., 1998*; *Lumini et al., 2010*; *Verbruggen & Toby, 2010*), and there are no comparative analyses on the AMF condition among different soil types under the same climate conditions in semi-arid regions.

Hitherto, traditional studies of AMF community composition have been scarce, partly due to the limitations of spore morphological features, which are easily influenced by external disturbances (*Oehl et al., 2004*), such as integrity of the spores (e.g., ability to identify spores). Due to the above defects, new research technologies are constantly updated. For instance, the development of molecular methods has greatly facilitated the studies of AMF taxonomic and phylogenetic reconstruction and has enhanced the sensitivity of AMF identification and quantification (*Lekberg et al., 2007*; *Helgason & Fitter, 2009*; *Balestrini et al., 2010*; *Gast et al., 2011*). Moreover, significant improvements have been made in the analysis of AMF condition by the high-throughput technology (*Margulies et al., 2006*). Determining the diversity of AMF became very widespread by using regions of the small ribosomal subunit gene. Due to technology advancements, it can provide the most comprehensive reference sequence data set (*Öpik et al., 2010*), and the sequencing data can provide detailed analyses on AMF communities among complex habitat types (*Öpik et al., 2013*). In summary, the application of new technologies will greatly improve the study of AMF communities.

Thus, our study applied the high-throughput sequencing (Illumina platform) to analyze the soil AMF communities in four habitat types, including agricultural arable land, artificial forest land, natural grassland, and bush/wood land, and in contrast to the first two soil habitat types, the last two types were undisturbed (without human interference). All habitat types were located in the Taihang Mountain, which belongs to the semi-arid ecosystem. We aimed to identify the relative importance of soil characteristics on AMF diversity and illustrate the differences in AMF communities among the predominant soil types. The

research would be a valuable contribution toward a better understanding on the way human activities have changed the composition of the current AMF communities, and the results would contribute to developeing a more precise guidance on local soil reclamation, vegetation restoration, and the maintenance of biodiversity in semi-arid regions.

## MATERIALS AND METHODS

### Study area

The research site was located in the south of Taihang Mountain ($112°28'$–$112°30'$E, $35°01'$–$35°03'$N), a site which belongs to the semi-arid area of China. The climate in the test area is temperate continental monsoon, with an annual average temperature of 14.3 °C and an average annual sunshine rate of 54%; the elevation gradient of our study sites ranged from 231 to 432 m above sea level. Soil in the study area is cinnamon (main part is similar to ustalf USDA), and the parent rock was composed mainly of sandstone and shale. The habitat types in this study were bush/wood land, forest land, grassland, and arable land. The bush/wood land included mainly *Vitex negundo* L, *Lespedeza bicolor* Turcz and *Ziziphus jujuba* Mill. var. *spinosa* (Bunge) Hu ex H.F. Chow, Forest land included mainly *Quercus variabilis* Bl., *Platycladus orientalis* (L.) Franco, and *Robinia pseudoacacia* L. Dominant herbaceous plants in the grassland were *Setaria viridis* (L.) Beauv., *Artemisia princeps* H. Lév. and Vaniot, *Pennisetum alopecuroides* (L.) Spreng., *Arthraxon hispidus* (Thunb.) Makino, and *Rehmannia glutinosa* (Gaetn.) Libosch. ex Fisch. et Mey. Finally, the prevalent herbaceous plants in the arable land were *Zea mays* L., *Triticum aestivum* L., *Ipomoea batatas* L., *Brassica campestris* L., and *Lycopersicon esculentum* Mill.

### Sample collection

In October 2016, soil samples were collected in triplicate at four sites (W1, BW, WL, and F). The sample collection occurred at the root zone of the plant at a soil depth of 5–10 cm (Table 1). Site W1 represented the forest land soil type; site BW had bush/wood soil type; site WL was characterized by grassland soil type; and arable land soil type was represented in site F. These 12 soil samples collected were placed in sterile plastic bags and transported in freezing boxes to the laboratory, and they were stored at $-70$ °C until further analysis.

### Soil geochemical analyses

We analyzed eight different soil factors, including soil pH, water content, available nitrogen ($NH_4^+$-N), available potassium ($K^+$-K) and phosphate phosphorus ($PO_4^{3-}$-P). Soil pH was examined by a pH meter (PX-KS06; Guangzhou Puxi Instrument, Guangzhou, China). Water content was measured by drying soil method, and the content of soil clay, silt, and sand was performed by using a Malvern Mastersizer (Mastersizer2000; Malvern Instruments, Malvern, UK). The available nitrogen and available potassium were analyzed by an Autoanalyzer (SEAL-AA3; SEAL Analytical, Milwaukee, WI, USA); phosphate phosphorus analyzed by $NaHCO_3$ Mo-Sb colorimetric method.
**Table 1** Geochemical characteristics of the soil samples and other information of the site of the present study.

| Sample sample | Soil type | Coordinates | pH | Water content (%) | Available nitrogen (mg kg⁻¹) | Available phosphorus (mg kg⁻¹) | Available potassium (mg kg⁻¹) | Clay (%) | Silt (%) | Sand (%) |
|---|---|---|---|---|---|---|---|---|---|---|
| W1-1 | | 35°1′56″N 112°29′1″E | 7.32 | 19.81 | 155.3 | 7.8 | 169.5 | 47.1 | 41.5 | 11.4 |
| W1-2 | Forest land | 35°2′16″N 112°28′20″E | 7.41 | 20.24 | 145.3 | 6.9 | 143.4 | 46.2 | 44.5 | 9.3 |
| W1-3 | | 35°2′45″N 112°28′52″E | 7.34 | 18.29 | 182.3 | 7.2 | 162.4 | 41.2 | 48.9 | 9.9 |
| Average | | | 7.36 A | 19.44 B | 161.0 A | 7.3 B | 158.4 B | 44.9 AB | 45.0 A | 10.2 C |
| BW-1 | | 35°1′49″N 112°29′14″E | 7.53 | 20.33 | 141.5 | 4.7 | 177.4 | 37.3 | 32.1 | 30.6 |
| BW-2 | Bush/ wood | 35°2′3″N 112°29′29″E | 7.41 | 21.14 | 133.2 | 4.8 | 186.4 | 41.2 | 31.5 | 27.3 |
| BW-3 | | 35°2′55″N 112°29′1″E | 7.36 | 23.00 | 187.3 | 6.5 | 160.1 | 28.3 | 42.9 | 28.8 |
| Average | | | 7.43 A | 21.49 B | 154 A | 5.3 B | 174.6 B | 35.6 BC | 35.5 B | 28.9 A |
| WL-1 | | 35°1′41″N 112°29′39″E | 7.21 | 16.71 | 167.4 | 4.3 | 129.1 | 33.2 | 52.6 | 14.2 |
| WL-2 | Grass land | 35°2′55″N 112°29′12″E | 7.34 | 15.90 | 144.3 | 5.1 | 135.4 | 26.4 | 49.6 | 24 |
| WL-3 | | 35°1′38″N 112°28′58″E | 7.42 | 18.05 | 132.2 | 2.4 | 122.0 | 37.5 | 48.9 | 13.6 |
| Average | | | 7.32 A | 16.88 C | 148.0 A | 3.93 B | 128.8 C | 32.37 C | 50.4 A | 17.3 B |
| F-1 | | 35°2′31″N 112°29′55″E | 7.27 | 26.71 | 177.5 | 18.7 | 287.4 | 55.2 | 39.5 | 5.3 |
| F-2 | Arable land | 35°2′12″N 112°29′38″E | 7.45 | 23.71 | 183.5 | 22.5 | 254.3 | 50.1 | 43.6 | 6.3 |
| F-3 | | 35°2′43″N 112°29′19″E | 7.33 | 23.95 | 182.3 | 30.2 | 299.1 | 47.8 | 47.5 | 4.7 |
| Average | | | 7.35 A | 24.79 A | 181.1 A | 23.8 A | 280.3 A | 51.0 A | 43.53 AB | 5.4 C |

**Notes.**
Values (eg. A, B, C) followed by the same letters in the same column was not significantly different ($p < 0.05$).

## Molecular analyses DNA extraction

A total of 50 mg soil was used for metagenomic DNA extraction in each sample, using the Fast DNA Isolation Kit (Q-BIOgene; Heidelberg, Germany). The extracts were stored at −20 °C for PCR. 1.0% agarose gels for checking DNA concentration and purity.

## Miseq sequencing step

Using the 18S rRNA gene and primer sets of AMV4.5N Forward 5′-AAGCTCGTAGTT-GAATTTCG-3′ and AMDG R 5′-CCCAACTATCCCTATTAATCAT-3′ to amplify the sequences (from soil DNA extracts), the primer had been reported to be acceptable in several previous studies (*Sato et al., 2005*). The initial PCR reactions were similar to the existing studies of (*Xiao et al., 2016*), including :25 μL total volumes, 1–2 μL DNA template, 250 mM dNTPs, 0.25 mM of primer, 1× reaction buffer and 0.5U Phusion DNA Polymerase.

The reactions used a 2720 model Thermal Cycler, and initial PCR amplification was conducted under the steps below: 94 °C for 2-min, then 25 cycles of 30-s denaturation at 94 °C, 30-s annealing at 56 °C, 30-s extension at 72 °C, 5-min extension at 72 °C.The second step PCR used a template, which come from the first 5uL product (without dilution). The second step PCR include: one cycle of 3-min at 94 °C, then 8 cycles of 30-s at 94 °C, 56 °C for 30-s and 72 °C for 30-s, and a 5-min extension at 72 °C. The PCR products were separated by electrophoresis (1.5% agarose gel in 0.5 × TBE) and purified using a gel xxtraction kit (Axygen Biosciences, Corning, NY, USA), then the libraries were

sequenced by PE300 sequencing on MiSeq v3 Reagent Kit (Illumina) platform (at Tiny Gene Company, Shanghai).

## Bioinformatics methods

The sequence reads were analyzed by the combination of software Mothur version 1.33.3, UPARSE (USEARCH version v8.1.1756) and R 3.2.2 (*R Development Core Team, 2015*), the original FASTQ files were demultiplexed through the barcode (*Schloss et al., 2009*). The PE reads for all samples were merged based on mothur. The low quality contigs were removed based on screen.seqs command by the settings filter (maxambig = 0, minlength = 200, maxlength = 580, the higher threshold can protect some longer sequences, which may be the correct fragment, maxhomop = 8). The decoded data information was aggregated (97% homology) to operational taxonomic units (OTUs) (*Edgar, 2013*).

BLAST analysis was conducted using the ''Nucleotide collection (nr/nt)'' database (https://blast.ncbi.nlm.nih.gov/Blast.cgi?PAGE_TYPE=BlastSearch). No threshold was set for *E* values, alignment length and identity settings. For each OTU representative sequence, a list of top BLAST hits was compiled. Uncultured clones were deleted from the list of top hits. The BLAST hit getting the highest score was identifed as the match's species.

## Statistical analyses

For the alpha-diversity analysis, Mothur version 1.33.3 software (*Schloss et al., 2009*) was used to analyze the OTU richness, Coverage, Chao, and Shannon's indices as reported earlier by *Schloss et al. (2009)*. The values of soil properties and diversity parameters were statistically analysed by SPSS V. 19 software (one-way ANOVA) (SPSS; IBM Corp., Armonk, NY, USA).

The clustering method was used with R v. 3.1.1 software to identify the AMF relationship (based on OTU abundance-based). Further, the indicator species analysis was utilized to identify the AMF communities associated with various habitat types (*Dufrene & Legendre, 1997*).

Using the Canoco software (Canoco for Windows 4.5 package) (*Braak & Smilauer, 2002*), we utilized Monte Carlo permutation and distance-based redundancy (db-RDA) tests to explain the correlation between soil AMF and geochemical factors. In addition, the heatmap results of the abundance percentages of AMF genera were obtained by Mothur version 1.33.3 software. The raw sequence information has been deposited into the NCBI database (accession number SRP116770).

## RESULTS

### Soil properties

For the eight geochemical factors measured, the arable land obtained the maximum values of water content, available phosphorus and available potassium (site F). Meanwhile, the minimum values of water content and available phosphorus were established in the grassland (site WL). In the bush/wood land (site BW), the maximum values of sand content (average 28.9%), but minimum silt content (35.5%) were established (Table 1).

**Table 2** The results of sequence data in the present study.

| | Soil type | Number of total sequence | Number of AMF OTUs | Coverage (%) | Chao's index | Shannon's index |
|---|---|---|---|---|---|---|
| W1-1 | | 26,866 | 62 | 99 | 67 | 2.93 |
| W1-2 | Forest land | 27,298 | 71 | 99 | 73 | 3.15 |
| W1-3 | | 28,493 | 60 | 99 | 60 | 2.53 |
| Average | | | | | 67 A | 2.87 B |
| BW-1 | | 31,391 | 73 | 99 | 74 | 3.46 |
| BW-2 | Bush/ wood | 35,206 | 69 | 99 | 70 | 3.38 |
| BW-3 | | 32,153 | 67 | 99 | 68 | 3.42 |
| Average | | | | | 70 A | 3.42 A |
| WL-1 | | 29,593 | 74 | 99 | 76 | 3.52 |
| WL-2 | Grass land | 28,148 | 83 | 99 | 87 | 3.49 |
| WL-3 | | 28,621 | 76 | 99 | 79 | 3.51 |
| Average | | | | | 81 A | 3.51 A |
| F-1 | | 18,900 | 52 | 99 | 67 | 3.46 |
| F-2 | Arable land | 19,135 | 62 | 99 | 88 | 3.38 |
| F-3 | | 15,095 | 54 | 99 | 55 | 3.45 |
| Average | | | | | 70 A | 3.43 A |

**Notes.**
The OTUs were defined at the cutoff 3% difference in sequence. Using the one-way analysis of variance (ANOVA) to evaluate statistical significance and results, followed by Tukey's HSD test. Capital direction symbols (eg. A, B, C) indicate full (5%) significance.

## AMF diversity data and community composition

In the current study, we have identified a total of 532,841 sequences and 803 OTUs from the total dataset; there were 320,899 sequences belonged to phylum Glomeromycotina (accounting for 60.2%). The number of sequences in each of the samples ranged from 15,095 to 35,206, and the number of AMF OTUs ranged from 52 to 83 (genetic distances of 3%). The OTUs' coverage in all soil types reached 99% (Table 2). On the basis of the OTU richness calculated by Chao's index, the grassland observed the greatest AMF value (site WL: 81). Through the analysis of Shannon's index, we discovered that the largest AMF diversity was also present in the grassland (site WL: 3.49–3.52 with an average value of 3.51), followed by the arable land (site F: 3.38–3.46 with an average value of 3.43), bush/wood land (site BW: 3.38–3.46 with average 3.42), and the forest land soils (site W1: 2.53–3.15 with an average value of 2.87) (Table 2).

Some variations in AMF community composition at the genus level were also detected among all soil samples. The 119 OTUs that could be classified were affiliated with ten AMF genera, whereas those that could not be identified were assigned as unclassified. The *Glomus* were the most abundant genera in all samples: 60%–75% in grassland, 70%–75% in arable land, 75%–80% in bush/wood, and 50%–70% in forest land. Meanwhile, their levels varied in the different soil types. *Ambispora* were found in all samples, but a greater abundance was detected in the grassland and arable land samples than in those of the bush/wood and forest land soils (Fig. 1).

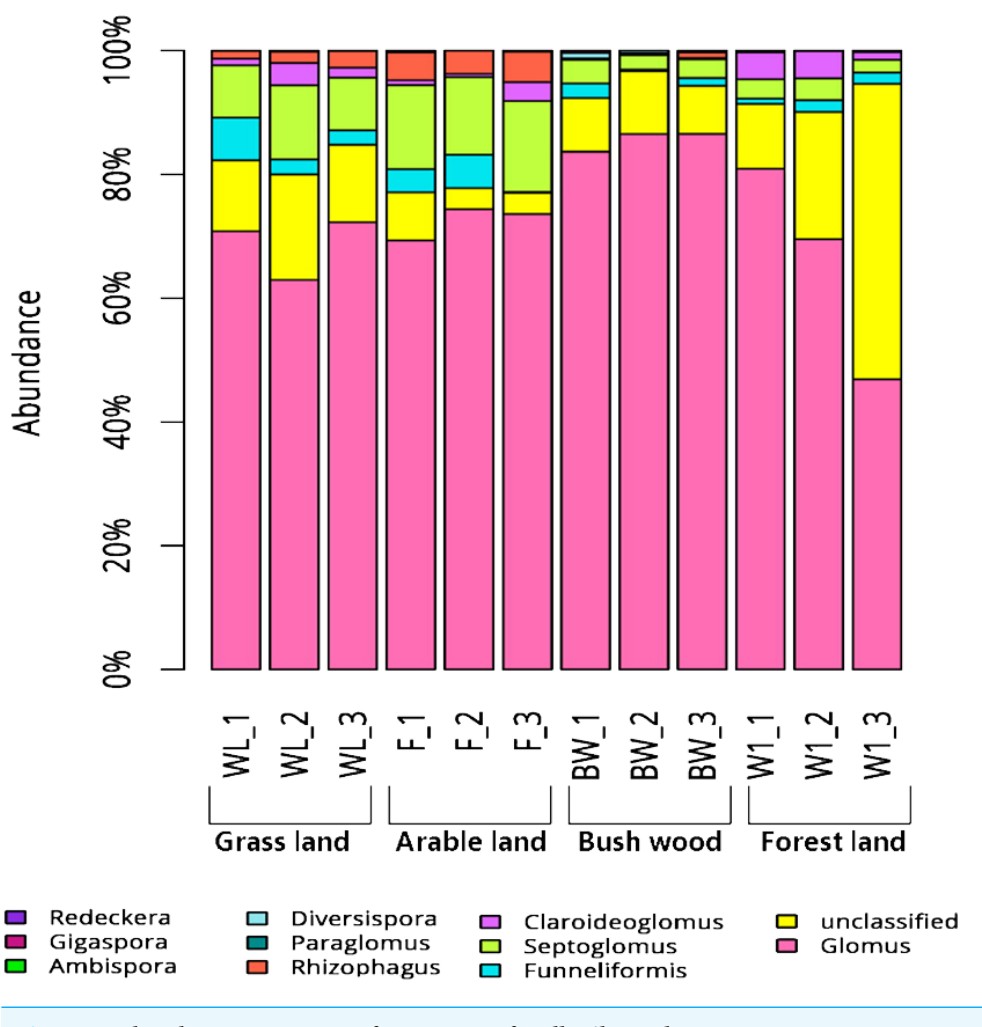

**Figure 1** Abundance percentages of AMF genera for all soil samples.

## Correlation among the three factors (AMF communities, soil types and environmental condition)

To determine the differences in soil AMF community, the OTU cluster analysis showed that the 12 soil samples were divided into four Soil Types (Fig. 2), and the indicator species analysis revealed that there were 60 AMF indicators (indicator value > 0.25, $p < 0.05$) in this four groups types, it mainly included bush/wood (*Glomus* and *Diversispora* taxa), arable land (*Glomus*, *Septoglomus* and *Rhizophagus* taxa), grassland (*Glomus* and *Septoglomus* taxa), forest land (*Glomus* and *Paraglomus* taxa) (Table S1).

The top 50 OTUs of all samples were selected and their abundances were compared by heatmap software. It revealed the relative distributions and abundances of the top 50 OTUs in all samples (Fig. 3). More detailed information about the top 50 OTUs was presented in Table S2. There is also a listing of all AMF OTUs and their closest matches in Table S3.

The distance-based redundancy analysis (db-RDA) showed that there was a significant correlation between the combination of eight environmental factors and soil AMF

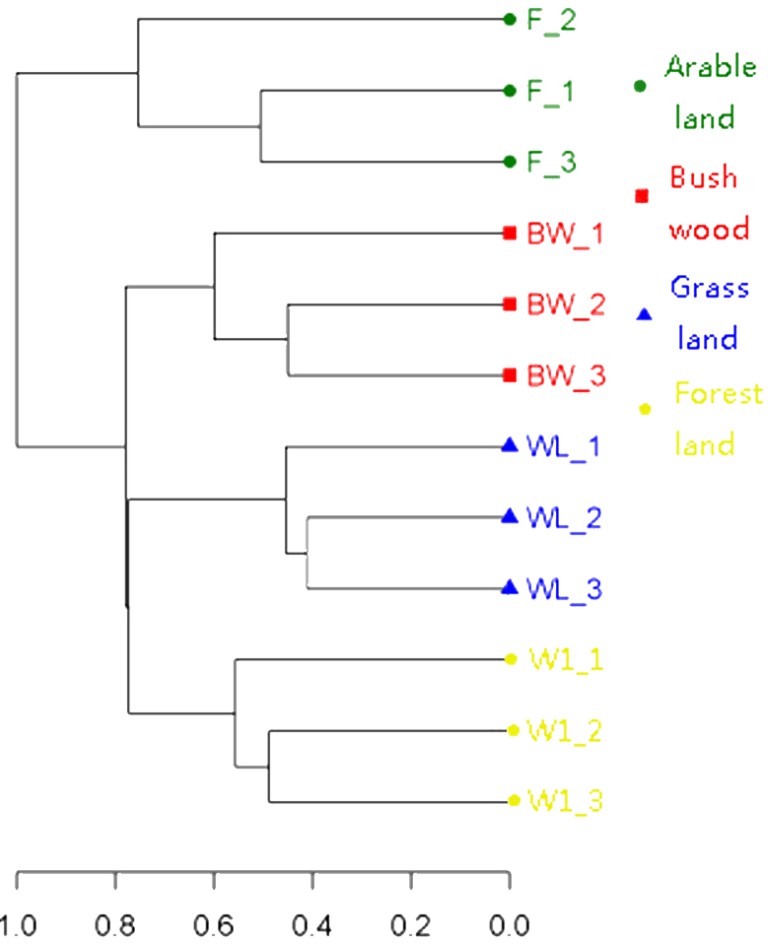

**Figure 2  Clustering analysis of AMF communities based on OTU abundance for each soil.**

community structure, that 81.9% of the soil community variation was attributed to all environmental factors (Fig. 4 and Table 3). However, using the Monte Carlo permutation test, we found that water content ($r^2 = 0.7332$, $p < 0.01$), available phosphorus ($r^2 = 0.7576$, $p < 0.01$), available potassium ($r^2 = 0.7973$, $p < 0.01$), silt ($r^2 = 0.6461$, $p < 0.05$), and sand ($r^2 = 0.6293$, $p < 0.05$) were important properties (Table 3).

## DISCUSSION

As mentioned earlier, the study area was located in the South Taihang Mountains of China, whose climate characterizes the region as a typical semi-arid climate zone. Under natural conditions, the thin soil layer, low forest coverage and much gravel are the characteristics of this area. Its forest types are mainly dominated by human intervention of *Quercus variabilis* Bl and *Platycladus orientalis* (L.); the vegetation is poor and only limited species could be planted (*Zhao, 2007*). Thus, improving local soil conditions and promoting plant growth are urgent tasks. However, some information had remained unexplored for the Taihang Mountain area, such as the distribution of AMF communities, the variation of

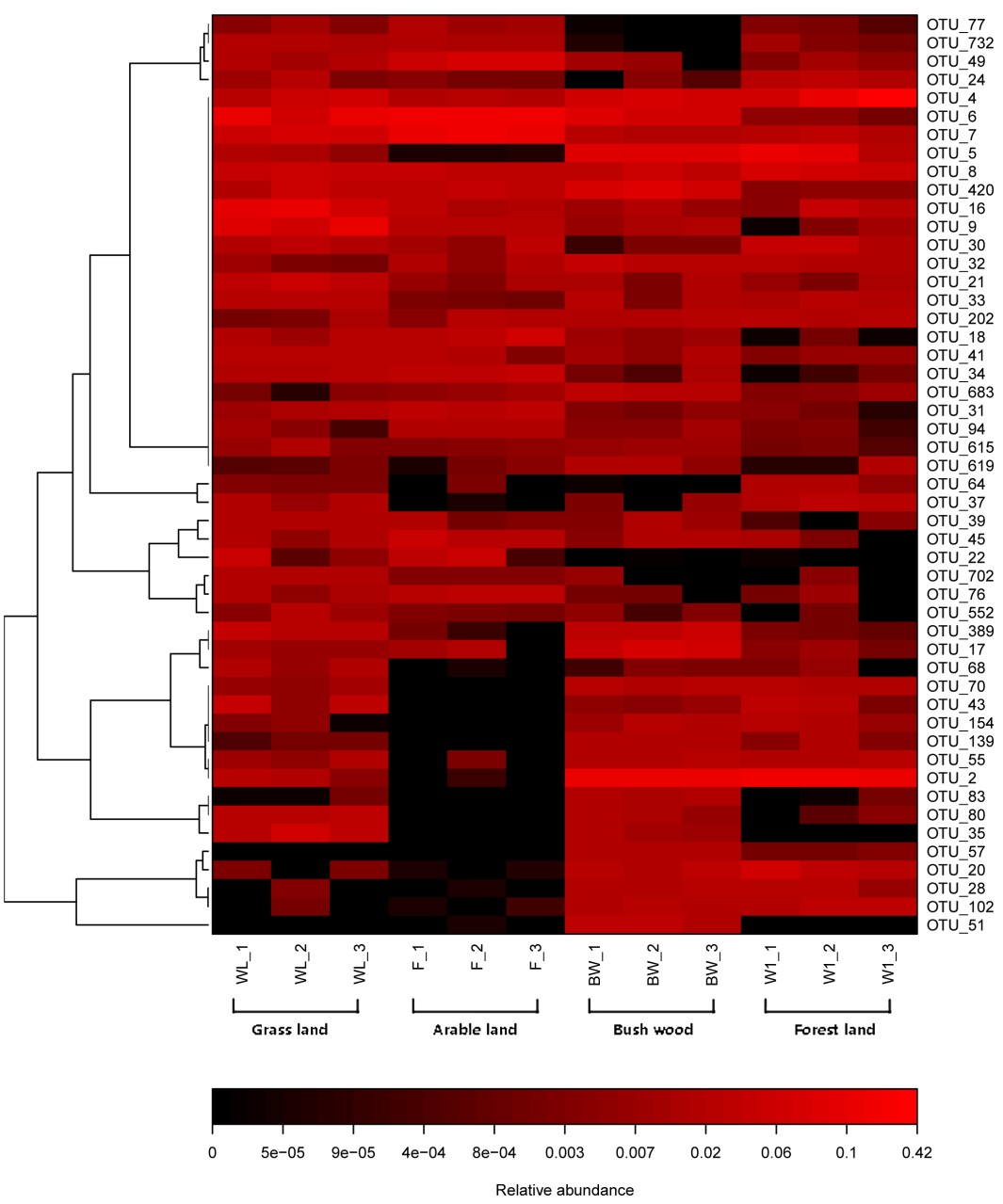

**Figure 3  Heat map of top 50 OTUs in all samples.** The color intensity (log scale) in each panel shows the percentage of a genus in a sample, referring to color key at the bottom.

AMF diversity, and the influence of various soil types on AMF composition. Therefore, in this study, we investigated the AMF communities among the predominant soil types in the South Taihang Mountain region. The results could be a valuable reference for improving the local ecological environment.

By analyzing the results of the four different soil types, the research showed that the diversity of AMF communities in undisturbed grassland soil type was greater than that in

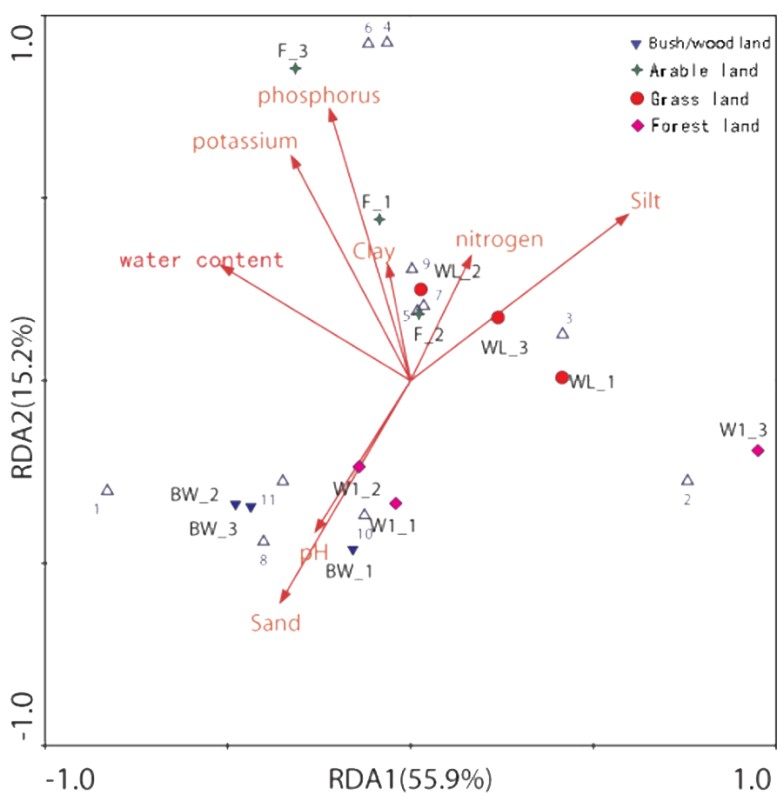

| | | 1 | Glomus |
| | | 2 | unclassified |
| | | 3 | Funneliformis |
| | | 4 | Septoglomus |
| | | 5 | Claroideoglomus |
| | | 6 | Rhizophagus |
| | | 7 | Paraglomus |
| | | 8 | Diversispora |
| | | 9 | Ambispora |
| | | 10 | Gigaspora |
| | | 11 | Redeckera |

**Figure 4** Distance-based redundancy (db-RDA) tests used to interpret the correlations between the AMF communities and environmental properties.

**Table 3** Monte Carlo permutation tests were used to detect the relationship between community composition and soil variables.

| | RDA1 | RDA2 | $r^2$ | $P$-value |
|---|---|---|---|---|
| pH | −0.2607 | −0.4144 | 0.2053 | 0.353 |
| Water content | −0.5188 | 0.3150 | 0.7332 | 0.004[**] |
| Available nitrogen | 0.1659 | 0.3414 | 0.3096 | 0.186 |
| Available phosphorus | −0.2218 | 0.7446 | 0.7576 | 0.004[**] |
| Available potassium | −0.3262 | 0.6153 | 0.7973 | 0.004[**] |
| Clay | −0.0622 | 0.3178 | 0.3611 | 0.156 |
| Silt | 0.5950 | 0.4549 | 0.6461 | 0.015[*] |
| Sand | −0.3564 | −0.6083 | 0.6293 | 0.013[*] |

Notes.
[*]Correlation is significant at the 0.05 level.
[**]Correlation is significant at the 0.01 level.
$P$-values based on 999 permutations.

artificial forest land (Table 2). That was consistent with *Öpik et al. (2008)*, who discovered that rich biological species composition and low external disturbance may lead to higher diversity of rhizosphere AMF of the natural vegetation soil. Our results showed that the value of Shannon's index in arable land was larger than that in artificial forest land. This outcome might have been caused by the cultivation practices implemented by the farmers, including the application of farmyard manure (food residues, livestock manure, etc.), which increased the number of microbial communities by raising the level of available nutrients (*Helgason & Fitter, 2009*). Indeed, it is generally accepted that the organic agriculture farming methods are regarded as a useful measure to increase AMF diversity (*Aroca, Porcel & Ruiz-Lozano, 2007*), and farmers in that region usually apply farmyard manure with cultivation methods that are closed to organic agriculture farming. On the other hand, probably because the growth and reproduction of specific AMF communities requiring particular host plant species, it leads to a less abundant community under a single artificial plantation habitat (*Long et al., 2010*). In general, human disturbance caused changes in the forest land environment, which reduced the transportation and distribution of AMF communities (*Yuan et al., 2008*), and the artificial forest land had the lowest AMF diversity in comparison with other land types.

Meanwhile, the results of the sequence data analysis of AMF community composition showed that members of both genera *Septoglomus* and *Glomus* existed in different soil types, including forest land, bush/wood, grassland, and arable land. Nevertheless, the representatives of *Glomus* were identified to be the main genus, and although *Glomus, Diversispora, Septoglomus, Rhizophagus* and *Paraglomus* were found in soils, only *Glomus* taxa served as indicator species for each habitat. These results are similar to previously published research that confirmed that the species of *Glomus* were the most abundant in the AMF assemblage (*Oehl et al., 2005*). The influence of certain factors may be the reason why *Glomus* was the dominant members in the AMF assemblage among those of other genera. Some researchers revealed that the species of *Glomus* genus can usually produce large numbers of spores and hypha fragments, which can colonize and extensively spread onto the roots of plants (*Öpik et al., 2006*). *Glomus* has also a certain resistance in complex environments (*Miransari et al., 2008*; *Bever et al., 2009*; *Barto et al., 2011*). Therefore, these features facilitate the survival and spread of *Glomus* genus members in a semi-arid mountain, and the emergence of this phenomenon is also the result of adaptation to the local ecological environment.

Moreover, our investigation established that water content is a significant factor which has an obvious effect on the AMF communities. Scholars have shown that the variations in the water content can contribute to changes in the physiological status of local AMF and its ecological niche directly (*Sieverding, Toro & Mosquera, 1989*), probably because the water was essential for the reproductive and metabolic processes. Thus, the water content can indirectly exert an impact on the distribution of AMF communities. In addition, our research also confirmed that there are significant relationships between the available phosphorus, available potassium, and soil AMF community structure. These interactions were most likely attributable to the fact that soil phosphorus may stimulate the spore germination and hyphal growth of AMF (*Miranda & Harris, 1994*), and the potassium

has the ability to increase the infection rate of AMF under drought stress (*Wei, 2016*). In general, soil nutrients can have on the growth of local AMF communities as the lack of nutrients inhibits the production and separation of spores (*Zaller, Frank & Drapela, 2011*). Thus, this work confirmed that environmental factors can drive the composition and distribution of AMF communities.

Furthermore, the composition of AMF communities seems to have been strongly influenced by the soil texture distribution, and our results showed that the content of silt and sand were significantly related to the soil AMF community (Table 1). The AMF diversity was higher in the samples from low-clay but high-sand content soil types. The appearance of the result was probably due to the fact that AMF is an aerobic organism, and the lower clay content provided better aeration, which was advantageous for plant root growth and soil humus decomposition, leading also to accelerated fungal propagation (*Torrecillas et al., 2014*). The research confirmed that AMF communities was negatively correlated with soil clay content.

## CONCLUSIONS

In conclusion, this study first delineated the species diversity and composition of AMF communities in Taihang Moutain, China. The members of the *Glomus* genus were predominant in all soil types. The findings also suggested that nutrient composition and soil texture were the most important factors affecting AMF communities. Moreover, there were differences in species diversity and composition of soil AMF communities among different habitat types. These findings shed new light on the characteristics of community structure and drivers of community assembly in AMF in semi-arid mountains, and point to the potential importance of different habitat types on AMF communities.

## ACKNOWLEDGEMENTS

The authors are grateful to the staff of the Xiaolangdi Ecological Station in Henan Province, China, for the provision of the soil materials and testing ground. We also would like to thank the Tiny Gene Bio-Tech (Shanghai) Co., Ltd. for their high-throughput sequence technology.

### Funding

This work was supported by the National Natural Science Foundation of China (number: 31270750). The funders had no role in study design, data collection and analysis, decision to publish, or preparation of the manuscript.

### Grant Disclosures

The following grant information was disclosed by the authors:
National Natural Science Foundation of China: 31270750.

## Competing Interests

The authors declare there are no competing interests.

## Author Contributions

- He Zhao performed the experiments, analyzed the data, wrote the paper, prepared figures and/or tables.
- Xuanzhen Li performed the experiments, wrote the paper, reviewed drafts of the paper.
- Zhiming Zhang contributed reagents/materials/analysis tools.
- Yong Zhao conceived and designed the experiments.
- Jiantao Yang contributed reagents/materials/analysis tools, prepared figures and/or tables.
- Yiwei Zhu performed the experiments.

## DNA Deposition

The following information was supplied regarding the deposition of DNA sequences:
    SRA: SRP116770.

## Data Availability

    The raw sequence information have been deposited into the NCBI database (accession number SRP116770).

## Supplemental Information

Supplemental information for this article can be found online at http://dx.doi.org/10.7717/peerj.4155#supplemental-information.

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
