# Peer review of "Species diversity and drivers of arbuscular mycorrhizal fungal communities in a semi-arid mountain in China"

_PeerJ, doi:10.7717/peerj.4155_

## Round 0.1 · original submission · Minor Revisions

Both reviewers appreciated your work and believe that it will make an important contribution to the scientific record. Both reviewers have minor comments for you to address. I have gone through and made additional comments to help you with the English (see attached PDF). Reviewer #2's requests regarding the discussion are optional. It is always better to discuss as much as you can, but it is not crucial for interpretation under the guidelines for the journal. If you can address these minor comments, the paper should be acceptable for publication.

·

Basic reporting

no comment

Experimental design

no comment

Validity of the findings

no comment

Additional comments

The authors explored the species diversity and composition of soil arbuscular mycorrhizal fungi communities in a semi-arid mountain – Taihang Mountain. They sampled soils from four habitat types, i.e., agricultural arable land, artificial forest land, natural grassland, and bush/wood land, and used the high-throughput sequencing with MiSeq for arbuscular mycorrhizal fungi. The sequencing depths are sufficient for arbuscular mycorrhizal fungi, and the statistical analyses are robust in detecting the important factors for arbuscular mycorrhizal fungal diversity and community composition. I only have minor comments to make it clearer.

Indicate “Taihang Mountain” in the abstract.
Did you sample the soil along elevations? As you mentioned the “mountain” in the title, please explicitly consider “elevation” or “temperature” as an explanatory variable in explaining diversity and community composition.
Figure 1, 2. Add soil types for replicated samples in figure legend or figure so that readers could understand the figure without referring to the main text.
In figure 3, taxonomic information for these 50 OTUs will enrich the information to readers.
Table 2. Make clear what kind of “results” you presented.
Table 1, 2. What the letters (A, B, C) mean?

Jianjun Wang
Nanjing Institute of Geography and Limnology, CAS

·

Basic reporting

The paper is well structured, overall the language is correct but several corrections lust be made. The literature cited is sufficient, recently updated and in accordance with the context of the paper. The pictures, figures and tables are correctly done and there was no overlapping of information. The discussion is mostly based on own experimental data, but further elaboration of own results and literature is needed regarding the effects of water content, and available P and K on AMF communities.

Experimental design

No comment

Validity of the findings

The discussion is mostly based on own experimental data, but further elaboration of own results and literature is needed regarding the effects of water content, and available P and K on AMF communities.

Additional comments

Further elaboration on the effects of most important environment factors on AMF community is needed, and some language corrections are needed. Some language corrections are suggested in the text.

---

## Round 0.2 · Minor Revisions

I have examined your edits and am requesting a few fixes to improve the use of English. I have included the line number and the text as it should be written.

L159. We used indicator species analysis to identify the AMF associated with different habitat types (Dufrene & Legendre P, 1997).

L 266. These interactions were present probably because phosphorus can significantly stimulate spore germination (Miranda & Harris, 1994), and potassium can increase the infection rate of rhizosphere fungi under drought stress (Wei, 2016)

L269. "In general", not "In genaral"

---

## Round 0.3 · accepted · Accept

Thank you for the additional revisions.